# Ionic Liquids as Organocatalysts for Nucleophilic Fluorination: Concepts and Perspectives

**DOI:** 10.3390/molecules27175702

**Published:** 2022-09-04

**Authors:** Young-Ho Oh, Dong Wook Kim, Sungyul Lee

**Affiliations:** 1Department of Applied Chemistry, Kyung Hee University, Duckyoung-daero 1732, Yongin City 446-701, Korea; 2Department of Chemistry, Inha University, 100 Inha-ro, Nam-gu, Incheon 402-751, Korea

**Keywords:** ionic liquids, organocatalysis, nucleophilic fluorination

## Abstract

Besides their extremely useful properties as solvent, ionic liquids (ILs) are now considered to be highly instructive tools for enhancing the rates of chemical reactions. The ionic nature of the IL anion and cation seems to be the origin of this fascinating function of ILs as organocatalyst/promoter through their strong Coulombic forces on other ionic species in the reaction and also through the formation of hydrogen bonds with various functional groups in substrates. It is now possible to tailor-make ILs for specific purposes as solvent/promoters in a variety of situations by carefully monitoring these interactions. Despite the enormous potentiality, it seems that the application of ILs as organocatalysts/promoters for chemical reactions have not been fully achieved so far. Herein, we review recent developments of ILs for promoting the nucleophilic reactions, focusing on fluorination. Various aspects of the processes, such as organocatalytic capability, reaction mechanisms and salt effects, are discussed.

## 1. Introduction

Ionic liquids (ILs) have found a wide range of applications in various fields of chemistry (organic, physical, medicinal, analytical and materials chemistry) because of their unique and useful physicochemical properties. In addition to being excellent solvents (due to high polarity) and environmentally benign solvents (because of low volatility, no metallic element, and easy recovery) ionic liquids are also gaining importance as catalysts/promoters in various chemical transformations [1,2,3,4,5,6,7,8,9,10,11,12,13,14,15,16]. Significantly enhanced reactivity and selectivity have also been observed in concerted reactions such as nucleophilic substitution reactions in ionic liquids [14,15,16]. By combining a variety of cations and anions with characteristic structures, it is possible to tailor-make task-specific ILs with desirable properties for specific chemical reactions [15,17,18,19,20,21,22,23,24,25,26,27,28].

Nucleophilic fluorination [13,14,16,29,30,31,32,33,34,35,36,37,38,39,40] has received considerable attention recently because of the importance of the synthesis of radiopharmaceutical substances to be used as diagnostic materials for the powerful technique of positron emission tomography (PET) [32]. Because the S_N_2 fluorination processes need not deal with the cumbersome F_2_ gas, they are advantageous over the electrophilic methods. There are two critical issues here, however, concerning the counter-cation and solvent (Figure 1). First, the common use of bulky counter-cations such as tetrabutylammonium (TBA^+^) may minimize the harmful Coulombic influence on the nucleophile, but usually suffers from a large amount of by-products (for example, E2 products). Alkali metal cations such as K^+^ or Cs^+^ may be more desirable, but these sources of fluoride exhibit the problem of solubility in organic solvents. Moreover, the strong electrostatic influence of alkali metal counter-cations was thought to decrease the reactivity of nucleophiles significantly. Second, protic solvents (water, alcohol), which are deemed to be favorable with regard to the solubility for alkali metal fluorides, were considered to be disadvantageous for nucleophilic reactions due to the hydrogen bonding between the –OH of solvent molecules and the nucleophile.

These two traditional ‘common senses’ for nucleophilic fluorination proved to be inaccurate in a series of recent investigations [13,14,16,32,33,34,35,36,37,38,39]: It was observed that the use of metal fluorides *in combination with* Lewis base promoters (oligoethylene glycol [33], *t*-butanol, *t*-amyl alcohol [38]) may lead excellent S_N_2 yields (>90%) in reasonable reaction time (<6 h). As the key to the mechanism of this class of phenomenally efficient and selective nucleophilic fluorination, it was proposed [32,33,36,37] that the solvents and/or promoters act as Lewis bases on the counter-cation, mitigating the unfavorable Coulombic influence of the latter on the nucleophile, thereby producing essentially ‘naked’ nucleophile.

In this respect, ILs were regarded as excellent solvent for metal fluorides, completely solving the problem of solubility of alkali metal fluorides (MFs). Due to the ionic nature of ILs, they can also be highly desirable promoters for nucleophilic fluorination using MFs. The IL cations and anions may facilitate the S_N_2 fluorination processes through their strong Coulombic interactions with MF, and also through the formation of hydrogen bonds with various functional groups in substrates.

Here, in this brief review, we describe the studies for IL-promoted nucleophilic fluorination that provided important advances in this very useful technology. Both experimental and theoretical (mechanistic) investigations are treated to encourage further developments. This review is not exhaustive, but discusses various features (efficacy of ILs, mechanism, salt effects, IL derivatives, etc.) of IL-facilitated fluorination.

## 2. Promotion of S_N_2 Fluorination by ILs Using Alkali Metal Fluorides

Kim, Song and Chi [14] reported in 2002 the first case of S_N_2 reactions using KF or CsF facilitated by ILs. They observed that for nucleophilic fluorination of 2-(3-methanesulfonyloxypropyl)naphthalene (Figure 2), the IL [bmim][BF_4_] tremendously increased the reaction rates. Using catalytic amounts (0.5 equiv) of [bmim][BF_4_] was enough to almost complete the reaction over 12 h with excellent S_N_2 yields (>85%) (Table 1). They also found that a small amount of water did not much affect the reaction, suggesting that the organocatalysis by ILs did not require anhydrous conditions. They also tried other ILs, finding similar results with the exception of [bmim][NTf_2_] in CH_3_CN (Table 2).

Studies for other metal fluorides by these authors showed that the reaction rates depended strongly on the alkali metal counter-cation [15]. For example, fluorination using the metal salts LiF and NaF did not proceed at all under the influence of [bmim][BF_4_] (Table 3). This observation is, of course, due to the difference in the Coulombic influence of the alkali metal counter-cations on the nucleophile (Li^+^ > Na^+^ > K^+^ > Rb^+^ > Cs^+^): The Coulombic attractive forces between the alkali metal cation Li^+^, Na^+^ and F^−^ are so strong [36] that the influence of the IL on Li^+^ and Na^+^ was not enough for facile reactions. In fact, to the best of our knowledge, the use of Li^+^ or Na^+^ as a counter-cation has not been successful so far for S_N_2 fluorination. Kim, Song and Chi applied this methodology to a variety of substrates, observing similar acceleration rates for a wide range of S_N_2 processes. These authors also proposed [13] that the role of ILs for enhancing the rates of nucleophilic substitution reactions was as an amphiphilic “electrophile-nucleophile” dual activator (acting both as Lewis acid on the nucleophile and as Lewis base on the counter-cation), which was the first such mechanistic insight to the ILs as organocatalysts for S_N_2 reactions. According to this conception, IL cation–IL anion–counter-cation–nucleophile–substrate form a compact configuration by Coulombic forces for facile S_N_2 reactions. The proof for this concept in molecular detail came in 2011, as described below.

## 3. S_N_2 Fluorination Facilitated by Polymer-Supported ILs

Considerable attention was also paid to polymer-supported (PS) ILs [16,41,42,43], which showed high efficacy as catalysts for nucleophilic fluorination. One of the advantages of this type of promoter/catalyst is, of course, the possibility of reuse. Kim and Chi [16] reported that polystyrene-based ILs [hmin]X (hmin = 1-n-hexyl-3-methylimidazolium cation, X = BF_4_, OTf) promoted fluorination reactions (such as that shown in Figure 3) in very high yields and can be reused for many S_N_2 reactions. Table 4 illustrates that fluorination reaction of the mesylate substrate (2) in the presence of 2.2 or 1.1 equiv of PS[hmim][BF_4_] was complete within 2 h, with excellent S_N_2 yields of 97–98% (entries 1 and 2), whereas the same reaction with 3 equiv of CsF in CH_3_CN at 100 C barely proceeded after 2 h (entry 5). Additionally, fluorination using PS[hmim][BF_4_] as an immobilized catalyst (entry 3) proceeded much faster than that using the same amount of IL as the catalytic system (entry 6). Chi and co-workers also studied [41] the synergistic effects of polymer-supported ionic liquid catalyst/*tert*-alcohol solvent system (Figure 2) for S_N_2 fluorination, in comparison with the corresponding reactions in *tert*-alcohol and in PS[hmim][BF_4_]/CH_3_CN. They observed the S_N_2 yield of 18, 40, and 84%, respectively, a significant increase due to the synergistic effects (Table 5). Application of this methodology to various substrates, leaving the groups –Br, –Cl, –OMs, –OTs and –OTf, was attempted, and Chi and co-workers mostly obtained > 80 % fluorination yields within 12 h.

## 4. Mechanistic Features of IL-Facilitated S_N_2 Fluorination Using Metal Fluorides

A rough inspection for catalysis of nucleophilic fluorination (and also other nucleophilic processes involving the nucleophiles other than F^−^) by ILs may look quite puzzling for two apparent reasons: First, the ionic liquid cation may slow the reactivity of F^−^ due to its strong Coulombic influence on F^−^. Second, the ionic liquid anion may try to stay as far away as possible from F^−^ also because of the repelling electrostatic interactions with F^−^. A clue to solving this difficulty of interpreting the experimental observed promotion by ILs came from the mechanism of S_N_2 reactions using alkali metal/nucleophile complexes facilitated by bulky protic solvents (*t*-butanol, *t*-amyl alcohol) and oligoethylene glycols. In contrast with conventional account of S_N_2 processes, the –OH groups in these promoters do *not* act as Lewis acids on F^−^ (thus, retarding the reactions), but as Lewis *bases* on the alkali metal counter-cation to alleviate the latter’s harmful electrostatic influence on F^−^, thereby producing an almost ‘naked’ nucleophile. This new mechanistic feature of S_N_2 processes was able to explain lots of experimental observations for the new class of S_N_2 reactions using KF or CsF in the presence of bulky alcohols and oligoethylene glycols. According to this mechanistic conception, an alternative feature of ionic interactions among the IL cation, IL anion, alkali metal cation (K^+^ or Cs^+^) and F^−^ would be envisioned: (1) the IL cation may not only interact with F^−^ but also with the IL anion, and thus these latter Coulombic interactions may significantly weaken the IL cation’s retarding influence on F^−^. (2) the IL anion, as Lewis bases, may act on K^+^ or Cs^+^ that mitigate the strong Coulombic influence of the alkali metal counter-cation, just as the O atoms of protic solvent molecules or of oligoethylene glycols. These strong interactions among the four ionic species may also help the formation of a pre-reaction complex and transition state that is highly favorable for S_N_2 attack of F^−^, hence the observed high S_N_2 selectivity.

Song and co-workers [13] discussed a range of chemical reactions to elucidate the origin of ‘positive effects of ionic liquids on catalysis’. The authors developed their concepts based on the ‘stabilization of highly reactive intermediates’ (vinyl cations, arenium cations, oxygen radical anions, etc.), formation of more reactive catalytic species for rare earth triflate catalyzed reactions, and ‘stabilized transition state’ for nucleophilic fluorination in ionic liquids. Song and co-workers’ conception of the IL-assisted S_N_2 fluorination (ILs act as amphiphilic “electrophile-nucleophile” dual activator: The counter-anion of ILs acts as a Lewis base toward K^+^, drastically reducing its electrostatic effects and thereby ‘freeing’ the nucleophile F^−^ and the acidic C2-proton interacts with the mesylate leaving group, helping it to detach from the reactant) was confirmed by Oh et al. [37] by quantum chemical calculations, treating the nucleophilic fluorination of 2-(3-methanesulfonyloxypropyl) naphthalene with CsF. Figure 1 depicts this mechanistic feature obtained by Oh et al. for fluorination reaction [Cs^+^F^−^ + C_3_H_7_OMs → C_3_H_7_F + Cs^+^OMs^−^] in [bmim][OMs]. The IL cation [bmim], IL anion [OMs], nucleophile F^−^, counter-cation Cs^+^ and the leaving group of the substrate form a very compact cyclic structure. The counter-cation Cs^+^ binds both to F^−^ and the leaving group -OMs, allowing an ideal configuration for the nucleophilic attack to be formed. The role of the IL cation [bmim] and IL anion [OMs] for accelerating the reaction is clearly seen: [OMs] (and the leaving group) interacts with the counter-cation Cs^+^, reducing the retarding Coulombic influence of Cs^+^ on F^−^ (thus “freeing” F^−^), whereas [bmim] ‘collects’ [OMs], Cs^+^ and F^−^ for this ideal configuration. The role of Cs^+^ promoting the approach of F^−^ to the leaving group (the substrate) and that of [OMs] (acting as a Lewis base) to neutralize the Coulombic influence of Cs^+^ on F^−^ seems to be the key factor in this mechanism. In this sense, the IL anion [OMs] plays its role just as the electronegative O atoms do in nucleophilic fluorination catalyzed by oligoethylene glycols [33] and bulky alcohols [38].

## 5. Salt Effects

Magnier and co-workers reported that, in their use [35] of the IL [bmim]F in solvent-free environments for S_N_2 fluorination, using 3 equiv. of the IL significantly enhanced the reaction yield as compared with the case of IL: substrate ratio = 1:1. This observation of enhanced fluorination yield from 47 to 84% by changing the IL: substrate molar ratio from 1:1 to 3:1 (Table 6) indicated significant salt effects that require mechanistic elucidation. In the process of quantum chemical analysis, Choi and Oh [44] found a new feature for the role of IL when the IL: substrate ratio = 2:1, as shown in Figure 2: The anion F^−^ in the second IL unit acts as an additional Lewis base on the two IL cations, not as a nucleophile, for facilitating the reaction. The calculated Gibbs free energy of activation decreases from 20.4 to 18.5 kcal/mol when the IL: substrate ratio increases from 1:1 to 2:1, in good agreement with the experimental observation of increased yield of fluorination.

A different kind of salt effect was observed by Grée and co-workers [45]. They found that adding KPF_6_^−^ or using 2 equiv. of KF in S_N_2 fluorination activated by [Bmin][PF_6_] significantly enhanced the rate of the reaction (Figure 3). These experimental observations were also analyzed by quantum chemical calculations by Oh and Lee [46] (Figure 4), with interpretations similar to those for Magnier and co-workers: The additional PF_6_^−^ or KF plays its role as an extra Lewis base acting on the counter-cation, further reducing the unfavorable influence of the latter on the nucleophile. Thus, the mechanistic study by quantum chemical methods was able to account for the experimentalists’ routine use of >2 equiv. metal salts for S_N_2 reactions.

## 6. Organocatalysis of Nucleophilic Fluorination by IL Derivatives

Attempts were made for improving the efficacy of ILs by synthesizing and using IL derivatives. Once the role of ILs as amphiphilic “electrophile-nucleophile” dual activators, and specifically the action of the IL anion as a Lewis base interacting with the alkali metal counter-cation, was elucidated, then the scheme for further development would be systematic and straightforward. Several investigators indeed carried out this task [34,47,48,49,50].

Shinde et al. [34] functionalized ILs with the *tert*-alcohol moiety, resulting in the 1-(2-hydroxy-2-methyl-n-propyl)-3-methylimidazolium mesylate IL ([C_1_im-tOH]-[OMs]), which was tested experimentally for the S_N_2 fluorination with CsF. Their results showed that the selectivity and yield increased considerably in comparison to other non-functionalized ILs, as an indication that the IL unit and *tert*-alcohol moiety produced synergetic effects. Oh et al. reported an activation barrier (Δ*G*^‡^) values of 20.8 kcal/mol when the reaction was carried out in the presence of the [C_4_mim][OMs] and 19.1 kcal/mol when the [C_1_im-tOH][OMs] was used, in agreement with the experimental yields of 32 and 100%, respectively (Table 7). Use of Oligoethylene glycolic imidazolium salts as a promoter for S_N_2 reactions by Kim and co-workers [47,48] is another new development toward tailor-making the ILs for specific purposes. The authors observed that these fused promoters exhibit much better efficacy for S_N_2 fluorination (Figure 5).

The mechanisms of the synergistic effects of these IL derivatives were treated elsewhere [49]; thus, here, we provide only a succinct description: The *t*-BuOH unit acts as an “anchor” to the leaving group for facile nucleophilic attack by F^−^, rather than as a Lewis base to Cs^+^, also helping to decrease the retarding effects of the H-F^−^ interactions. On the other hand, the six O atoms of the two hexaethylene glycol units in [dihexaEGmim][X] ILs coordinate K^+^, and the two –OH groups, each from the two hexaEGs, act as an “anchor” to the nucleophile F^−^, thereby helping the formation of a ‘free’ nucleophile that is essentially separated from the counter-cation.

## 7. Aromatic (S_N_Ar) Fluorination of Diaryliodonium Salts Using MFs

Aromatic (S_N_Ar) [51] fluorination has profound implications with regard to the synthesis of aromatic radiopharmaceuticals that may be used as diagnostic or therapeutic agents for PET technology. One typical example would be the synthesis of [^18^F]F-dopa. This class of nucleophilic fluorination has been under intensive study since the development of diaryliodonium salts as very useful substrates by Pike and coworkers [52] in 1998. The efficacy of solvent/promoter ILs for fluorination of diaryliodonium salts [52,53,54,55,56,57,58,59,60,61,62,63,64,65,66], however, has not been systematically tested so far. There seems to be a great possibility that ILs could make a profound difference in the efficiency of this class of reaction, because the hypervalent I^+^ in the substrates may directly interact with the IL anion by the strong Coulombic force. In contrast with organic solvent, the counter-anion to I^+^ and the counteraction to F^−^ may also interact with the IL cation/anion in a very complicated way. For S_N_Ar radiofluorination, the use of weak bases such as K_2_CO_3_ is necessitated to extract ^18^F^−^, which could further complicate the nature of interactions. Thus, mechanistic investigation for ‘cold’ fluorination (introduction of ^19^F) would be considerably easier. Discussion of (S_N_Ar) fluorination, including metal-catalyzed processes, would require a separate review; thus, we only give a brief description of some recent developments in the organocatalysis of diaryliodonium salts using MFs, in order to stimulate experimental and theoretical studies for this promising subject.

Figure 4 depicts the cold and hot S_N_Ar fluorination reactions first advanced by Pike and co-workers [52], in which the hot reaction reached 80% completion after 40 min in CH_3_CN. This method proved a powerful technique to fluorinate the electron-*rich* rings with improved yield and controlled regioselectivity that had been difficult to achieve by conventional methods.

This nucleophilic protocol developed by Pike and co-workers was employed by Wirth and co-workers [58] for the multi-step synthesis of the elusive radiotracer [^18^F]F-dopa using chiral phase-transfer catalysts (Figure 5), which was further advanced later to the one-step synthetic technique [59]. The RCY was reported to be >35 %. A number of advantages (for example, no need to handle carrier-added [^18^F]F2 gas) of the nucleophilic procedure involving diaryliodonium salts over the electrophilic [^18^F]-fluorination were demonstrated. Figure 4 shows multistep synthesis to [^18^F]6-fluoro-3,4-dihydroxy-L-phenylalanine ([^18^F]F-dopa) using chiral phase-transfer catalysts.

A more recent synthetic route for [^18^F]F-DOPA adopted by Maisonial-Besset et al. [61] seems to be an advance, because their method obviates the need for the use of base, cryptand or metal catalyst with improved RCY (27–38%). The fully automated synthetic procedure starts from the *t*-butyl ester precursor (Figure 6).

6-[18F]fluorodopamine ([^18^F]F-DA) is another radiotracer for imaging neuroendocrine tumors. Snyder and co-workers’ [62] one-pot synthetic scheme (Figure 7) is a notable development. The diaryliodonium precursor leads to the intermediate after initial ion exchange at the hypervalent iodine center, and deprotection of the radiolabeled intermediate in HCl produces [^18^F]F-DA. A radiochemically pure product was achieved with a radiochemical yield of 12%.

As noted above, the use of ILs as the solvent/promoter for these S_N_Ar fluorination processes, to the best of our knowledge, has not been attempted so far. It would be extremely intriguing to examine this possibility both experimentally and theoretically.

## 8. Theoretical Approaches: Supramolecule–Continuum vs. QM/MM Theory

Several theoretical methods have been employed to treat the promotion of nucleophilic fluorination in ILs. Since the ILs are in a liquid state, however, some simplifying models should have been adopted. One choice was the supramolecule (cluster)–continuum approach, in which the IL molecules directly interacting with the substrate/metal fluoride are treated by full quantum chemical methods (mostly based on density functional theory (DFT)), whereas the rest of the infinite number of IL molecules are as continuum. The biggest merit of this approach is simplicity, and it is good for obtaining and comparing the relative Gibbs free energies of alternative reaction paths, providing the essential features of the reaction processes. Pliego [39,67] employed the molecular dynamics simulation combined with (cluster)–continuum approach to treat bulky alcohols as reaction medium for S_N_2 nucleophilic fluorination, specifically for the solvation of alkali metal salts and the transition states in *tert*-butanol solution. The author found that the transition state was solvated by a cluster of four *tert*-butanol molecules and by a dielectric continuum (Figure 6). The free energy of activation obtained by cluster–continuum calculations for CsF reaction with ethyl bromide was 28.4 kcal/mol, close to the experimental value of 28.9 kcal/mol for a similar system. This technique also seems to be promising for S_N_2 reactions in ILs for calculating and comparing the Gibbs free energies of activation to estimate the efficacy of promoter/catalysts and also to compare the relative feasibility of alternative reaction pathways, as demonstrated by Lee and co-workers in a series of works [32,36,37]. For the supramolecule (cluster) part of the system, it would be prerequisite to accurately calculate the weak interactions (hydrogen bonding, specifically). Thus, DFT-based methods with such capability such as M06-2X [68] would be desirable. A number of polarizable continuum methods (PCM) may be used for the treatment of solvent continuum, but the SMD method [69] was quite adequate in terms of computational cost and accuracy.

An alternative model would be the mixed quantum and molecular mechanical (QM/MM) simulation method in which the IL molecules in indirect interaction with the QM-treated part of the system are treated by MM methodology. Recent excellent simulations by Gallo and co-workers [70] are to be noticed in this regard. Figure 7 depicts the structures of the reactant, TS, and product for S_N_2 fluorination in [Bmim][Br]. On average, five IL units were calculated to surround the contact ion-pair KF. Thus, Gallo and co-workers’ QM/MM simulation clearly illustrates that the metal salt KF reacts as a contact ion pair nucleophile in [Bmim][Br], in line with Lee and co-worker’s theory [37]. The calculated Gibbs free energy of activation ΔG^‡^ of 22~24 kcal/mol [70] for the reactions in five IL solvents are comparable to those obtained by the supramolecular–continuum approach.

A series of QM/MM simulations for reactions in ILs by Acevedo and co-workers [71,72,73,74,75,76] also exhibits the potentiality of the technique for IL-facilitated chemical transformations. Table 8 shows that the free energies of activation obtained by the semiempirical QM method PDDG/PM3 and force field for the S_N_Ar reactions of piperidine and 2-methoxy-5-nitrothiophene are in better agreement [71] with experimental values [77] than those calculated by the supramolecular–continuum approach (B3LYP/6-311++G(2d,p)/PCM). Figure 8 illustrates the first-solvation shell of the S_N_Ar substrates (piperidine, 2-methoxy-5-nitrothiophene) encapsulated by [bmim][BF_4_] ILs, which may not be obtained by the supramolecule–continuum approach. Improvement may also be achieved by employing Born–Oppenheimer molecular dynamics (MD) calculations by Acevedo and co-workers [72] for the structures of [Bmin]X (X = Cl, BF_4_, PF_6_) ionic liquids, and H_2_ evolution from formic acid decomposition in an ionic liquid solvent by Klein and co-workers [78]

## 9. Conclusions

Use of ILs for facilitating chemical reactions has been and will be much wider than that described in this review. Since the aliphatic fluorination is just a small part of S_N_2 reactions, many advantageous properties of ILs will surely stimulate further developments in other nucleophilic reactions with a variety of substrates and organocatalysts. S_N_Ar reactions seems to remain a largely unexplored field in this respect [79,80,81,82]. Experimental and theoretical examination of using ILs as solvents/promoters for efficient and selective S_N_Ar fluorination in metal-free conditions would be highly desirable. Understanding the underlying mechanism by quantum chemical calculations will certainly help to tailor-make the ILs for aliphatic and aromatic fluorination. Development of IL derivatives containing a reinforcing (via interactions with counter-cation and/or with the leaving group) side-chain is an avenue to be further explored. Use of ILs for synthesizing ^18^F-labeled radiophamaceutical substances also looks promising. Although not an IL, the recently reported reaction of ^18^F-fluorination by a *tert*-butanol-integrated quaternary ammonium salt (tri-(*tert*-butanol)-methylammonium fluoride [83,84] shows this possibility.

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
