# Peer review of "Ionic Liquids as Organocatalysts for Nucleophilic Fluorination: Concepts and Perspectives"

_molecules, 2022, doi:10.3390/molecules27175702_

Round 1

Reviewer 1 Report

This manuscript is interesting and could be accepted for publication in Molecules. The topic is very relevant for both modern organic chemistry and catalysis.

Besides their extremely useful properties as solvent, ionic liquids (ILs) are now considered to be highly instructive tools for enhancing the rates of chemical reactions. The ionic nature of the IL anion and cation seems to be the origin of this fascinating function of ILs as organocatalyst/promoter through their strong Coulombic forces on other ionic species in the reaction and also through the formation of hydrogen bonds with various functional groups in substrates. It is now possible to tailor-make ILs for specific purposes as solvent/promoters in a variety of situations by carefully monitoring these interactions. Despite the enormous potentiality, it seems that the application of ILs as organocatalyst/promoter for chemical reactions have not been fully achieved so far. Herein authors review recent developments of ILs for promoting the SN2 reactions, focusing on nucleophilic fluorination. Various aspects of the processes, such as organocatalytic capability, reaction mechanisms and salt effects, are discussed.

The manuscript provide sufficient background in the topic of ionic liquids as organocatalysts for nucleophilic fluorination. The topic is definitely original. This review add some understanding of the mechanisms and driving factors of discussed processes to the subject area compared with other previously published material. The manuscript is good illustrated and interesting to read. The style is fine (may be, very minor polishing of English language could be recommended). The conclusions consistent with the evidence and arguments presented. Overall, authors address the main question posed. I have only couple of minor suggestions.

- Following relevant references focused on noncovalent organocatalysts should be briefly cited in introduction: J. Org. Chem. 2022, In press. DOI: 10.1021/acs.joc.2c00680; J. Org. Chem. 2022, In press. DOI: 10.1021/acs.joc.2c01141; J. Org. Chem. 2022, V. 87. P. 4569.; Org. Biomol. Chem. 2021, V. 19. P. 7611.; RSC Adv. 2021, V. 11. P. 4574.

- Some more detailed perspectives regarding the future research could be formulated in conclusions section.

Author Response

Reviewer 1:

We thank the Reviewer for many helpful comments.

-Following relevant references focused on noncovalent organocatalysts should be briefly cited in introduction: J. Org. Chem. 2022, In press. DOI: 10.1021/acs.joc.2c00680; J. Org. Chem. 2022, In press. DOI: 10.1021/acs.joc.2c01141; J. Org. Chem. 2022, V. 87. P. 4569.; Org. Biomol. Chem. 2021, V. 19. P. 7611.; RSC Adv. 2021, V. 11. P. 4574.

We add the papers as Ref. 28, 65 - 66

-Some more detailed perspectives regarding the future research could be formulated in conclusions section.

We revise the Conclusion

from:

Use of ILs for facilitating chemical reactions has been and will be much wider than that described in this review. Since the aliphatic fluorination is just a small part of SN2 reactions, many advantageous properties of ILs will surely stimulate further developments in other nucleophilic reactions with a variety of substrates and nucleophiles. SNAr reactions seems to remain as a largely unexplored field in this respect [58–62]. Development of IL derivatives containing a reinforcing (via interactions with counter-cation and/or with the leaving group) side-chain is an avenue to be further explored. Use of ILs for synthesizing 18F-labeled radiophamaceutical substances also looks promising. Although not a IL, recently reported reaction of 18F-fluorination by a tert-butanol-integrated quaternary ammonium salt (tri-(tert-butanol)-methylammonium fluoride [63,64] shows this possibility.

to:

Use of ILs for facilitating chemical reactions has been and will be much wider than that described in this review. Since the aliphatic fluorination is just a small part of SN2 reactions, many advantageous properties of ILs will surely stimulate further developments in other nucleophilic reactions with a variety of substrates and organocatalysts. SNAr reactions seems to remain a largely unexplored field in this respect [58–62]. Experimental and theoretical examination of using ILs as solvent/promoter for efficient and selective SNAr fluorination in metal-free conditions would be highly desirable. Understanding the underlying mechanism by quantum chemical calculations will certainly help to tailor-make the ILs for aliphatic and aromatic fluorination. Development of IL derivatives containing a reinforcing (via interactions with counter-cation and/or with the leaving group) side-chain is an avenue to be further explored. Use of ILs for synthesizing 18F-labeled radiophamaceutical substances also looks promising. Although not a IL, recently reported reaction of 18F-fluorination by a tert-butanol-integrated quaternary ammonium salt (tri-(tert-butanol)-methylammonium fluoride [63,64] shows this possibility.

Reviewer 2 Report

The topic of this publication is quite interesting, but this is not review. This is report of authors own research. The idea of review is to present for researchers the wider overview for the entitled topic. Only in the introduction there are some information about the other work but without visualization (schemes or figures.). There is only presentation of authors results , that’s why I cannot accept this as review. It can be a good perspective or feature article of authors work. After the change of type of article I will be able to accept with minor changes.

Author Response

Reviewer 2:

We thank the Reviewer for many helpful comments.

-The topic of this publication is quite interesting, but this is not review. This is report of authors own research. The idea of review is to present for researchers the wider overview for the entitled topic. Only in the introduction there are some information about the other work but without visualization (schemes or figures.). There is only presentation of authors results, that’s why I cannot accept this as review. It can be a good perspective or feature article of authors work. After the change of type of article I will be able to accept with minor changes.

We fully appreciate the Reviewer’s concern. Originally we planned to the subject of organocatalysis by ILs for all chemical reactions, including nucleophilic substitution reactions other than fluorination, but then the scope seemed to be too wide. Basically we had in mind a Tutorial Review (Chem. Soc. Rev.) type review paper, which focuses mainly on the authors’ own research along with other labs’ works. We therefore decided to on <nucleophilic fluorination in ILs> only to cope with several constraints (space and time). As for this topic, Kim, Song and Chi’s work (J. Am. Chem. Soc. 2002, 124, 10278–10279) was a seminal one, and subsequent studies naturally followed. We didn’t discuss our own works only, however. Experimental investigations in other labs that led to our theoretical works were fully treated, and theoretical studies by Acevedo, Pliegos, and other works were also included. One thing we did not treat properly may be the aromatic nucleophilic substitution (SNAr) fluorination. Although study for SNAr fluorination in ILs is scarce, we add a new Section <7. Aromatic (SNAr) fluorination of diaryliodonium salts using MFs> to discuss the recently developed methodology of aromatic fluorination of diaryliodonium salts, with regard to the stimulation for the corresponding studies of SNAr fluorination in ILs. We also added following references.

Neumann, C. N.; Hooker, J. M.; Ritter, T., Concerted nucleophilic aromatic substitution with 19F and 18F, Nature 2016, 534, 369–373.

Bortolami, M,; Mattiello, M,; Scarano, V.; Vetica, F.; Feroci, M., In Situ Anodically Oxidized BMIm-BF4: A Safe and Recyclable BF3 Source, J. Org. Chem. 2021, 86, 16151–16157.

Beil, S.; Markiewicz, M.; Pereira, C. S.; Stepnowski, P.; Thöming, J.; Stolte, S., Toward the Proactive Design of Sustainable Chemicals: Ionic Liquids as a Prime Example, Chem. Rev. 2021, 121, 13132-13173.

Zhang, X.; Lu, G.-p.; Xu, Z-.b. Cai, C. Facile Synthesis of Indolizines via 1,3-Dipolar Cycloadditions in [Omim]Br: The Promotion of the Reaction through Noncovalent Interactions, ACS Sustainable Chem. Eng. 2017, 5, 9279–9285.

Sysoeva, A. A.; Novikov, A. S.; Il'in, M. V.; Suslonov, V. V.; Bolotin, D. S., Predicting the catalytic activity of azolium-based halogen bond donors: an experimentally-verified theoretical study, Org. Biomol. Chem. 2021, 19, 7611-7620.

Shah, A.; Pike, V.W.; Widdowson, D.A. The synthesis of [18F]fluoroarenes from the reaction of cyclotron-produced [18F]fluoride ion with diaryliodonium salts. J. Chem. Soc. - Perkin Trans. 1 1998, 2043–2046.

Yoshimura, A.; Zhdankin, V. V., Advances in Synthetic Applications of Hypervalent Iodine Compounds, Chem. Rev. 2016, 116, 3328-3435.

Yagyu, T.;Takemoto, Y.; Yoshimura, A.; Zhdankin, V. V.; Saito, A., Iodine(III)-Catalyzed Formal [2 + 2 + 1] Cycloaddition Reaction for Metal-Free Construction of Oxazoles, Org. Lett. 2017, 19, 2506-2509.

Antonkin, N. S.; Vlasenko, Y. A.; Yoshimura, A.; Smirnov, V. I.; Borodina, T. N.; Zhdankin, V. V.; Yusubov, M. S.; Shafir, A.; Postnikov, P. S., Preparation and Synthetic Applicability of Imidazole-Containing Cyclic Iodonium Salts, J. Org. Chem. 2021, 86, 7163-7178.

Xu, B.; Gao, Y.; Han, J.; Xing, Z.; Zhao, S.; Zhang, Z.; Ren, R.; Wang, L., Hypervalent Iodine(III)-Mediated Tosyloxylation of 4-Hydroxycoumarins, J. Org. Chem. 2019, 84, 10136–10144.

Chen, H.; Wang, L.; Han, J., Deacetylative Aryl Migration of Diaryliodonium Salts with C(sp2)–N Bond Formation toward ortho-Iodo N-Aryl Sulfonamides, Org. Lett. 2020, 22, 3581–3585.

Edwards, R.; Wirth, T., [18F]6-fluoro-3,4-dihydroxy-L-phenylalanine – recent modern syntheses for an elusive radiotracer, J. Label Compd. Radiopharm 2015, 58. 183–187.

Edwards, R.; Westwell, A.D.; Daniels, S.; Wirth, T. Convenient synthesis of diaryliodonium salts for the production of[18F]F-DOPA. European J. Org. Chem. 2015, 58, 625–630.

Tian, T.; Zhong, W.-H.; Meng, S.; Meng, X.-B.; and Li, Z.-J., Iodine Mediated para-Selective Fluorination of Anilides, J. Org. Chem. 2013, 78, 728–732.

Maisonial-Besset, A.; Serre, A.; Ouadi, A.; Schmitt, S.; Canitrot, D.; Leal, F.; Miot-Noirault, E.; Brasse, D.; Marchand,P.; Chezal, J.-M., Base/Cryptand/Metal-Free Automated Nucleophilic Radiofluorination of [18F]FDOPA from Iodonium Salts: Importance of Hydrogen Carbonate Counterion, Eur. J. Org. Chem. 2018, 48, 7058–7065.

Vāvere, A. L.; Neumann, K. D.; Butch, E. R.; Hu, B.; DiMagno, S. G.; Snyder, S. E., Improved, one‐pot synthesis of 6‐[18F]fluorodopamine and quality control testing for use in patients with neuroblastoma, J. Label Compd. Radiopharm. 2018, 61, 1069–1080.

Jang, K. S., Jung, Y.-W.; Gu, G.; Koeppe, R. A.; Sherman, P. S.; Quesada, C. A.; Raffel, D. M., 4‑[18F]Fluoro‑m‑hydroxyphenethylguanidine: A Radiopharmaceutical for Quantifying Regional Cardiac Sympathetic Nerve Density with Positron Emission Tomography, J. Med. Chem. 2013, 56, 7312−7323.

Han, J.; Chen, H.; An, G.; Sun, X.; Li, X.; Liu, Y.; Zhao, S.; Wang, L., Hypervalent Iodonium Zwitterions and Nucleophilic Aromatic Substitution: A Multiple-Step Experiment in Organic Chemistry, J. Chem. Educ. 2021, 98, 3992–3998.

Il’in, M. V.; Sysoeva, A. A,; Novikov, A. S.; Bolotin, D. S., Diaryliodoniums as Hybrid Hydrogen- and Halogen-Bond-Donating Organocatalysts for the Groebke–Blackburn–Bienaymé Reaction, J. Org. Chem.  2022, 87, 4569-4579.

Yunusova, S. N.; Novikov, A. S.; Soldatova, N. S.; Vovk, M. A.; Bolotin, D. S., Iodonium salts as efficient iodine(III)-based noncovalent organocatalysts for Knorr-type reactions, RSC Adv. 2021, 11. 4574-4583.

Round 2

Reviewer 2 Report

This publication is interesting, but I am still confused that this is not review but only report of authors own research, besides of addition of isotopic research. However, after the present improvement I accept this manuscript, but still in the introduction there should be some visualization (schemes or figures) of cited works. Without that introduction is very poor. 

Author Response

We thank the Reviewer for many helpful comments.

-This publication is interesting, but I am still confused that this is not review but only report of authors own research, besides of addition of isotopic research. However, after the present improvement I accept this manuscript, but still in the introduction there should be some visualization (schemes or figures) of cited works. Without that introduction is very poor. 

We added Scheme 1 (pp. 2) to illustrate examples studying the effects of counter-cation and solvent/promoter on nucleophilic substitution reactions from Ref. [35] and [40], respectively.
